# 13-Decyl Berberine Derivative Is a Novel Mitochondria-Targeted Antioxidant and a Potent Inhibitor of Ferroptosis

**DOI:** 10.3390/cells14241963

**Published:** 2025-12-10

**Authors:** He Huan, Alisa A. Panteleeva, Ruben A. Simonyan, Armine V. Avetisyan, Natalia V. Sumbatyan, Konstantin G. Lyamzaev, Boris V. Chernyak

**Affiliations:** 1Belozersky Institute of Physico-Chemical Biology, Lomonosov Moscow State University, 119992 Moscow, Russia; 2Faculty of Bioengineering and Bioinformatics, Lomonosov Moscow State University, 119234 Moscow, Russia; 3Chemistry Department, Lomonosov Moscow State University, 119991 Moscow, Russia; 4The “Russian Clinical Research Center for Gerontology” of the Ministry of Healthcare of the Russian Federation, Pirogov Russian National Research Medical University, 129226 Moscow, Russia

**Keywords:** berberine derivatives, mitochondria-targeted antioxidants, ferroptosis, mitochondrial lipid peroxidation, drug development

## Abstract

Berberine is a plant isoquinoline alkaloid widely used in traditional medicine for the therapy of diabetes, cardiovascular and other diseases. Ferroptosis, a regulated form of cell death driven by lipid peroxidation, is thought to contribute to the pathogenesis of various diseases associated with excessive oxidative stress. The therapeutic actions of berberine are mediated, at least in part, by its antioxidant effects. Here, we report that the lipophilic berberine derivative 13-decyl berberine (C10Berb) is a mitochondria-targeted antioxidant that exhibits superior ferroptosis inhibition compared to native berberine in H9c2 cardiomyocytes and human fibroblasts. C10Berb efficiently accumulates in mitochondria, suppressing both mitochondrial lipid peroxidation, reactive oxygen species formation, and lipofuscin accumulation at concentrations markedly lower than berberine. Mechanistic studies indicate that the anti-ferroptotic effect of C10Berb is independent of AMPK or Nrf2 activation and is primarily due to its direct antioxidant activity in mitochondria. In isolated cardiac mitochondria, C10Berb potently inhibited lipid peroxidation induced by either reactive oxygen species produced in the electron transport chain or artificial free radical initiators. These results support the hypothesis that mitochondrial lipid peroxidation is critical for ferroptosis and highlight the potential of mitochondria-targeted berberine derivatives as promising therapeutic agents for conditions associated with ferroptotic cell death.

## 1. Introduction

Berberine is an isoquinoline alkaloid present in various plant species of the *Berberidaceae*, *Papaveraceae*, and *Ranunculaceae* families. It is widely used in traditional Chinese medicine for the therapy of diabetes and inflammatory disorders. The medicinal use of berberine as the main bioactive component of *Rhizoma coptidis* dates back to 200 AD [1]. Recent studies have demonstrated therapeutic potential of berberine against diabetes [2] and injury to multiple organs exposed to oxidative stress [3,4,5,6] including myocardial ischemia/reperfusion [7].

One of the best-performed clinical trials of berberine was conducted in 100 subjects with fatty liver disease and type 2 diabetes [8]. Patients receiving 1000 mg of berberine (as a salt of ursodeoxycholic acid) twice a day for 18 weeks were reported to experience significant reductions in liver fat, improved glycemic control, decreased serum liver enzyme levels, and significant weight loss [8]. Very high doses used in this trial presumably reflects poor oral absorption and low bioavailability of berberine [3]. Intravenous administration improves the bioavailability of berberine, but can cause serious side effects, including drop in blood pressure and respiratory arrest [9]. Various types of nanocarriers such as liposomes [10], phytosomes (phospholipid micelles) [11], or chitosan-coated lipid nanoparticles [12] have been proposed to improve the bioavailability of berberine, but the problem remains unresolved.

Despite numerous studies (nearly 9000, according to PubMed), the mechanisms of berberine’s therapeutic action remain unclear. This appears to reflect its ability to modulate a plethora of important cellular signaling pathways including AMPK/mTOR, Nrf2, MAPK, PI3K/Akt, etc. Berberine activates the AMP-activated protein kinase (AMPK), which regulates general metabolism and activates autophagy through inhibitory phosphorylation of mTOR [13]. AMPK also activates SIRT1 (sirtuin 1), a member of NAD+-dependent histone deacetylase family, targeting expression of important defense proteins controlled by the FOXO (Forkhead Box O) transcription factors (i.e., FOXO1 and FOXO3) [14]. FOXO enhances the expression of genes encoding antioxidant enzymes, including catalase, superoxide dismutase (SOD), and glutathione peroxidases [15]. Activation of autophagy and simultaneous stimulation of mitochondrial biogenesis controlled by PGC-1α (peroxisome proliferator activated receptor γ-coactivator 1α) in AMPK/SIRT1/FOXO pathway results in improvement of mitochondrial quality control as an important component of antioxidant defense [16].

Another protective pathway activated by berberine is mediated by the transcription factor Nrf2 (nuclear factor erythroid-2 related factor-2), which is involved in the cellular response to various stresses and xenobiotics [17]. In addition, Nrf2 can be activated by AMPK and this has been reported to be important for the anti-inflammatory effect of berberine [18]. Nrf2 controls the expression of important antioxidant enzymes NQO-1(NADPH quinone oxidoreductase-1) and HO-1 (heme oxygenase-1) [19], thus berberine may induce a wide range of antioxidant mechanisms. Moreover, berberine exhibits direct free radical scavenging activity [20,21], while its role in cellular models and in vivo remains unclear. The antioxidant effect of berberine may be amplified by its ability to chelate transition metal ions such as iron and copper [22].

Ferroptosis is a type of regulated non-apoptotic cell death dependent on iron-catalyzed lipid peroxidation [23]. An increasing number of studies indicate that ferroptosis makes a significant contribution to the pathogenesis of diseases associated with ischemia/reperfusion or iron overloaded of various tissues, neurodegenerative and autoimmune diseases, acute kidney injury, and many other pathologies [24,25]. Berberine has been used in several models of pathologies putatively associated with ferroptosis. In a spinal cord injury model, berberine reduced neuronal ferroptosis, presumably via activation of the AMPK-NRF2-HO-1 pathway, thereby enhancing cellular resistance to oxidative stress [26]. At the same time, berberine was reported to attenuate cerebral ischemia–reperfusion injury associated with ferroptosis due to modulation of gut microbiota, and antibiotic administration counteracted its protective effect [27]. Only very limited studies have examined the anti-ferroptosis effects of berberine in well-controlled cell models. One of these studies suggested that berberine inhibits ferroptosis of mouse hippocampal neuronal cells HT22 via Nrf2 activation [28]. Berberine has been shown also to inhibit ferroptosis induced by erastin (a plasma membrane cystine-glutamate antiporter inhibitor) or RSL3 (glutathione peroxidase 4 inhibitor) in cardiomyocytes [29], although the mechanism of the effect remains unknown.

Mitochondria play an important role in ferroptosis [30]. Our recent studies in several cell models have shown that mitochondrial lipid peroxidation is critical for ferroptosis [31,32,33]. Berberine has a positive charge, so it can accumulate in the mitochondria due to the negative membrane potential inside. However, experimentally, accumulation of berberine in the mitochondria of living cells was observed only at very high concentrations [34]. In our earlier study, we conjugated berberine (and the related cationic alkaloid palmatine) by aliphatic (decyl) linker with the antioxidant plastoquinol moiety to deliver the antioxidant to the mitochondria [35,36]. These conjugates (SkQBerb, SkQPalm) as well as their plastoquinol-free analogs penetrated through planar phospholipid bilayer membrane in their cationic forms and accumulated in the mitochondria of living cells. In the current study, we examined the antiferroptotic potential of decyl berberine (C10Berb) in H9c2 cardiomyocytes and human fibroblasts. It was shown that C10Berb suppressed erastin- or RSL3-induced ferroptosis at significantly lower concentrations than berberine. This effect was due to the selective accumulation of C10Berb in mitochondria and the inhibition of mitochondrial lipid peroxidation.

## 2. Materials and Methods

### 2.1. Chemicals

SkQ1 (10-(6-plastoquinonyl)decyltriphenylphosphonium), SkQBerb (13-[9-(4,5-dimethyl-3,6-dioxocyclohexa-1,4-dien-1-yl)nonyloxycarbonyl-methyl]berberine) and C10Berb (13-(decyloxycarbonyl-methyl)berberine) were kindly provided by the Institute of Mitoengineering, Lomonosov Moscow State University. C11-BODIPY581/591 was from Lumiprobe (Moscow, Russia). CM-H2DCFDA and MitoSOX were from Invitrogen Life Technologies (Waltham, MA, USA). MitoCLox was synthesized from succinimidyl ester of C11-BODIPY581/591 and (5-[(4-aminobutyl)amino]-5-oxopentyl) triphenylphosphonium bromide as described in [37]. Other reagents, except for those indicated, were from Sigma-Aldrich (Saint Louis, MO, USA).

### 2.2. Cell Culture

The rat cardiomyoblast cell line H9c2 (EcACC, Cat. No. 88092904) and primary human dermal fibroblasts (Research Facility “Biobank,” Research Centre for Medical Genetics, Moscow, Russia) were cultured in 4.5 g/L glucose Dulbecco’s Modified Eagle’s Medium (DMEM) (Gibco; Thermo Fisher Scientific, Inc., Waltham, MA, USA) supplemented with 2 mM L-glutamine, 10% fetal bovine serum (FBS) (HyClone, Logan, UT, USA), and 100 U/mL penicillin and 100 U/mL streptomycin (Gibco, USA).

Cell viability was determined using the CellTiter-Blue^®^ assay (Promega, Madison, WI, USA) following the manufacturer’s protocol, and fluorescence was recorded at 560 nm excitation and 590 nm emission using a Fluoroskan Ascent FL reader (Thermo Labsystems, Waltham, MA, USA). Background fluorescence from wells containing medium and reagent only was subtracted, and viability was expressed as a percentage of untreated control.

### 2.3. Fluorescence Microscopy

Cells were plated in 35 mm glass bottom dishes for confocal microscopy (SPL Life science, №100350) at 100,000 cells. After incubation with C10Berb 100 nM for 24 h cells were washed by PBS twice and stained with 100 nM MitoTracker Red for 15 min. Fluorescence was analyzed using a fluorescence microscope Olympus IX 83 (Tokyo, Japan) in GFP channel.

### 2.4. Detection of Mitochondrial Lipid Peroxidation Level and Lipofuscin

Cells were incubated with 5 μM erastin or 100 nM RSL3 for 18 h in the presence of 200 nM MitoCLox. After incubation, cells were detached using trypsin–EDTA solution, centrifuged (900× *g*, 5 min, 4 °C), resuspended in 50 μL PBS, and analyzed using an Amnis FlowSight imaging flow cytometer (Luminex Corporation, Seattle, WA, USA). Fluorescence was excited at 488 nm and detected in channels 505–560 nm (Ch2) and 595–642 nm (Ch4). The degree of MitoCLox oxidation was assessed from the Ch2/Ch4 fluorescence ratio. Spectral compensation was performed to correct fluorescence spillover between detection channels and to minimize interference from the intrinsic fluorescence of C10Berb. Channel 2 (Ch2) was used to detect autofluorescence, which indicates lipofuscin content Data were processed using Amnis IDEAS^®^ 6.2 software (Luminex, Seattle, WA, USA). For each sample, at least 4000 events were recorded. In cases of significant deviation from a Gaussian distribution, a gating procedure was applied for accurate data evaluation.

### 2.5. Detection of Mitochondrial and Cytosolic ROS

Cytosolic ROS levels were measured using 5-(and-6)-chloromethyl-2′,7′-dichlorodihydrofluorescein diacetate (CM-H_2_DCF-DA) (Invitrogen Life Technologies, Waltham, MA, USA), while mitochondrial ROS were detected using MitoSOX (Invitrogen Life Technologies). Cells were incubated with 5 μM erastin or 100 nM RSL3 for 18 h, then the medium was replaced with fresh medium containing 1.8 μM CM-H_2_DCF-DA or 1 μM MitoSOX, and incubation was continued for 30 min. Fluorescence was measured using Amnis FlowSight cytometer (Luminex Corporation) at 488 nm excitation in channel Ch2 (505–560 nm) for CM-H_2_DCF-DA or Ch3 (560–595) for MitoSOX.

### 2.6. Isolated Mitochondria

Mitochondria were isolated from rat hearts in a medium containing 250 mM sucrose, 5 mM MOPS-KOH (pH 7.4), 1 mM EGTA, and 0.5 mg/mL bovine serum albumin as described in [38]. Cardiac tissue was minced in ice-cold isolation medium (10 mL/g tissue) and homogenized in a Potter glass homogenizer for 1–2 min. The homogenate was diluted to 20 mL/g tissue and centrifuged for 10 min at 600× *g*. The supernatant was collected and centrifuged for 10 min at 12,000× *g*. The pellet was resuspended in a minimal volume, re-homogenized, diluted in 20 mL isolation medium, and centrifuged again (10 min, 12,000× *g*). The final pellet was resuspended and stored on ice.

Mitochondria were incubated for the indicated time in medium containing 250 mM sucrose, 5 mM MOPS-KOH (pH 7.4), 50 μM FeSO_4_, 5 mM glutamate, 5 mM malate, and 10 μM rotenone at 37 °C with stirring. Protein concentration was 0.4 mg/mL (Bradford assay). For C10Berb fluorescence measurement, mitochondria were lysed in 1% SDS, and spectra were recorded between 370 and 600 nm upon 365 nm excitation using a FluoroMax-3 spectrofluorometer (Horiba Scientific, Kyoto, Japan). MitoCLox oxidation was assessed as described previously [32].

### 2.7. Statistics

Results are presented as the mean ± standard deviation (SD) of at least three independent experiments. Comparisons were made using one-way ANOVA. Statistical significance was assessed with Prism 10.0 software (GraphPad Software, LLC, Solana Beach, CA, USA); values of **** *p* ≤ 0.0001; *** *p* ≤ 0.001; ** 0.001 < *p* ≤ 0.05; * *p* < 0.05 were considered significant.

## 3. Results

### 3.1. C10Berb Is More Effective in Protecting Against Ferroptosis than Berberine

Inhibition of ferroptosis by berberine and its analogs was studied in H9c2 rat cardiomyocytes and in human dermal fibroblasts (HDF) (Figure 1). Ferroptosis was induced using either erastin—an inhibitor of the cystine-glutamate antiporter system xc^−^—or RSL3, a glutathione peroxidase 4 (GPX4) inhibitor [39]. In both cell models, these inhibitors induced necrotic cell death as evidenced by propidium iodide nuclear staining, which was not prevented by the pan-caspase inhibitor zVAD-fmk. This excludes secondary caspase-dependent necrosis. Cell death was completely suppressed by the ferroptosis inhibitor ferrostatin-1 and the iron chelator deferoxamine (DFO), confirming that ferroptosis was the mode of cell death (Figure 1b,d).

As previously reported, erastin-induced ferroptosis is effectively blocked by the mitochondria-targeted antioxidant SkQ1 and by methylene blue (MB), which prevents the formation of reactive oxygen species (ROS) in mitochondria [31,32,33,40]. As shown in Figure 1b,d, both SkQ1 and MB also inhibit RSL3-induced ferroptosis to a similar extent, reinforcing the role of mitochondrial ROS in ferroptosis triggered by GPX4 inhibition.

Importantly, the data in Figure 1 demonstrate that C10Berb is significantly more effective in preventing ferroptosis than berberine in both H9c2 and HDF models. Notably, SkQBerb, a C10Berb analog carrying an antioxidant plastoquinol moiety (Figure 1a), was only slightly more effective than C10Berb (Figure 1c,e). This limited enhancement suggests a lack of synergy between the berberine and plastoquinol moieties in SkQBerb, implying that both compounds may act on the same mitochondrial targets.

We previously showed that both C10Berb and SkQBerb penetrate an artificial phospholipid membrane in their cationic forms and accumulate selectively in the mitochondria of HeLa carcinoma cells [35,36]. Here, we extended these findings to H9c2 cardiomyocytes and HDF, comparing the uptake and subcellular localization of C10Berb and berberine. As expected, C10Berb, which is more hydrophobic, accumulated in cells more efficiently than berberine, presumably due to faster penetration and binding to cell membranes (Figure 2a). The kinetics of C10Berb accumulation in H9c2 cardiomyocytes and HDF, measured after solubilization of cells with 1% SDS, did not differ significantly (Figure 2b). Fluorescence microscopy showed selective accumulation of C10Berb in the mitochondria of both cell types (Figure 2c). Berberine accumulation was detected only at higher concentrations (10 μM) and was characterized by a diffuse cytoplasmic distribution (Appendix A). Consistent with previous reports [41], berberine also accumulates in mitochondria of HepG2 cells, but only at higher concentrations (5–30 μM) substantially above the levels required for ferroptosis protection. These findings indicate that the hydrophobic decyl chain in C10Berb significantly enhances membrane permeability and mitochondrial accumulation, resulting in markedly improved protection against ferroptosis.

### 3.2. C10Berb Inhibits Ferroptosis by Preventing Mitochondrial Lipid Peroxidation

In vivo studies have suggested that berberine’s therapeutic effects may involve activation of AMP-dependent protein kinase (AMPK) and transcription factor Nrf2 signaling pathways [26]. Activation of AMPK by C10Berb may be mediated by efficient inhibition of complex I in the mitochondrial electron transport chain, as shown previously [35,36]. The same mechanism is responsible for AMPK activation by berberine [42]. We examined the potential activation of AMPK and Nrf2 by C10Berb and berberine in fibroblasts (Appendix A). In these experiments, we did not observe significant accumulation of the Thr172-phosphorylated activated form of AMPK (p-AMPK) by either C10Berb (100 nM) or berberine (10 μM). These data contradict numerous reports of AMPK activation by berberine, and we have no explanation for this discrepancy. To analyze the potential activation of Nrf2, we measured Nrf2 protein accumulation (Western blotting) and heme oxygenase-1 expression (RT-PCR). The expression of these genes is typically activated by Nrf2. Berberine (10 μM), but not C10Berb (100 nM), was shown to significantly activate Nrf2.

To test the possible inhibition of ferroptosis by AMPK- and Nrf2-dependent mechanisms, we used several known inducers of these pathways. AMPK was stimulated by the specific inhibitor of Complex I rotenone, the adenosine analogue 5-aminoimidazole-4-carboxamide riboside (AICAR), and the glycolysis inhibitor 2-deoxyglucose (2DG), whereas Nrf2 was stimulated by sulforaphane (Sulf) [43]. The data in Figure 3 show that neither of these inducers protects against erastin- or RSL3-induced ferroptosis. Thus, we conclude that neither AMPK nor Nrf2 is responsible for the inhibition of ferroptosis by C10Berb.

To analyze the antioxidant action of C10Berb, we measured mitochondrial lipid peroxidation (mitoLPO) using the mitochondria-targeted fluorescent ratiometric probe MitoCLox [37]. Selective accumulation of MitoCLox in mitochondria and its specific oxidation by mitochondrial LPO-generated lipid radicals have been previously shown [44]. In our previous studies, this probe was used to demonstrate that mitoLPO precedes ferroptosis induced by erastin or the glutathione biosynthesis inhibitor buthionine sulfoximine in fibroblasts, as well as by exogenous iron in H9c2 cells [32]. Here, we modified the MitoCLox application protocol by increasing the incubation time with the probe. These changes significantly improve the sensitivity and reproducibility of the assay. As shown in Figure 4, both erastin and RSL3 induced substantial MitoCLox oxidation, confirming the activation of mitoLPO during ferroptosis. Since RSL3-induced MitoCLox oxidation was observed here for the first time, we tested the effects of two compounds preventing ROS accumulation in mitochondria, SkQ1 and MB, in this model. Both SkQ1 and MB prevented MitoCLox oxidation (Figure 4) in parallel with the inhibition of RSL3-induced ferroptosis (Figure 1b,d) indicating that mitoLPO is critical for ferroptosis induced by glutathione peroxidase 4 inhibition.

C10Berb also prevented MitoCLox oxidation at the same concentration that inhibited ferroptosis (Figure 4) and SkQBerb had a similar effect. Berberine (Berb) prevented MitoCLox oxidation only at 100 times higher concentrations. Overall, these data indicate that C10Berb is an effective mitochondria-targeted antioxidant and inhibition of mitoLPO plays an important role in its protective effect against ferroptosis.

To further investigate the antioxidant effect of C10Berb, we measured mitochondrial superoxide anion production using MitoSOX and cytosolic ROS accumulation using CM- H_2_DCF DA (Figure 5). Both erastin and RSL3 significantly increased mitochondrial and cytosolic ROS levels in H9c2 and fibroblasts. The mitochondria-targeting agents SkQ1 and MB prevented the accumulation of mitochondrial as well as cytosolic ROS, indicating that mitochondrial ROS production is critical for the induction of overall oxidative stress during ferroptosis. C10Berb and SkQBerb replicated these effects, strongly reducing both mitochondrial and cytosolic ROS (Figure 5). In all cases, berberine (Berb) reduced mitochondrial and cytosolic ROS only at concentrations 100 times higher than C10Berb and SkQBerb.

It is known that during the aging of an organism or cells in culture, they accumulate colored aggregates of lipofuscin (LF), which consists mainly of highly oxidized proteins and a smaller amount of lipids, as well as carbohydrates and nucleic acids linked by numerous covalent cross-links [45]. It is assumed that mitochondria serve as one of the main sources of LF [46]. Lipid peroxidation is an important factor in the accumulation of LF, and iron chelators that block lipid peroxidation prevent the accumulation of LF both in vitro [47] and in vivo [48]. Recently, we have shown that exogenous iron added in the form of iron(III)-ammonium citrate causes a rapid (24 h) accumulation of LF in H9c2 cells, which precedes ferroptosis [32]. As can be seen from Figure 5, erastin and RSL3 also caused accumulation of LF, which was detected by characteristic fluorescence in the region of 450–550 nm. C10Berb, as well as DFO, SkQ1 and MB blocked both mitoLPO and accumulation of LF, which indicated an important role of mitoLPO in the formation of LF. These findings provide further evidence that C10Berb protects cells from ferroptosis by suppressing mitochondrial oxidative stress at its origin.

We cannot exclude the possibility that ROS formed in the mitochondria participate in the induction of ferroptosis along with the products of lipid peroxidation. At the same time, one of the inducers of ferroptosis formation in our experiments was RSL3, an inhibitor of the mechanism that specifically protects lipids from oxidation. If the action of numerous antioxidant mechanisms could be disrupted under the action of erastin, which suppresses the biosynthesis of glutathione, then the inhibition of glutathione peroxidase 4 selectively stimulated lipid peroxidation without affecting other components of the cell. Moreover, the iron chelator DFO, which selectively blocks LPO, prevented the accumulation of ROS (Figure 4 and Figure 5), confirming the conclusion about the accumulation of ROS in cells as a result of the development of LPO during ferroptosis.

### 3.3. C10Berb Inhibits Lipid Peroxidation in Isolated Mitochondria

In our previous study [31], we showed that lipid peroxidation detected by MitoCLox in isolated rat heart mitochondria could be induced by the complex I inhibitor rotenone in the presence of iron ions (Fe^2+^) and NAD-dependent respiratory substrates (glutamate and malate). SkQ1 and MB were shown to inhibit LPO in isolated mitochondria. Using this assay, we showed that C10Berb also inhibited LPO (Figure 6a). To exclude possible inhibition of mitochondrial ROS production, induced LPO in mitochondria using a lipophilic free radical initiator, 2,2′-azobis(2,4-dimethylvaleronitrile) (AMVN) in the presence of iron ions and respiratory substrates. C10Berb inhibited LPO in this assay at the same concentrations as in the case of rotenone-induced LPO (Figure 6b) SkQBerb inhibited LPO at the same concentrations. Comparison with the standard antioxidant trolox showed that the antioxidant efficacy of C10Berb in isolated mitochondria was 10–20 times higher.

The direct antioxidant effect of berberine and its derivatives depends on its redox state. Reduced forms of berberine, i.e., dihydroberberine and tetrahydroberberine, are also natural alkaloids with strong antioxidant properties [49]. In our previous study, we analyzed the effect of C10Berb on LPO in artificial liposomes and showed that it inhibited LPO only after reduction with borohydride [35]. It is known that berberine can be reduced by bacteria of the gut microbiota [50]; however, the possible reduction of berberine in animal cells or in isolated mitochondria has not been studied. We analyzed the possible reduction of C10Berb in cardiac mitochondria using the loss of berberine fluorescence upon reduction. Incubation of C10Berb with mitochondria for 2 h in the presence of rotenone and respiratory substrates resulted in a marked decrease in fluorescence at 550 nm (excitation at 365 nm), indicating reduction of the berberine residue (Figure 6c). Prolonged incubation (24 h) led to an almost complete loss of fluorescence, suggesting extensive reduction over time. The low overall rate of reduction may reflect rapid autoxidation of the reduced C10Berb forms within the mitochondrial environment.

## 4. Discussion

As shown in Figure 1, C10Berb inhibited erastin- or RSL3-induced ferroptosis more effectively than berberine in both H9c2 cardiomyocytes and human fibroblasts. This superior efficacy is likely due to the significantly greater mitochondrial accumulation of C10Berb in these cells (Figure 2). We previously demonstrated that mitoLPO plays a critical role in ferroptosis triggered by erastin or buthionine sulfoximine-induced glutathione depletion [31]. In the present study, we extended this observation to RSL3-induced ferroptosis, which is driven by inhibition of glutathione peroxidase 4 and stimulates LPO independently of intracellular glutathione levels. We found that RSL3-induced ferroptosis is preceded by mitoLPO as detected using the mitochondria-targeted fluorescent probe MitoCLox [37,44]. C10Berb suppressed both ferroptosis and mitoLPO at comparable concentrations (Figure 4). It also reduced ROS accumulation in the mitochondrial matrix, as measured by MitoSOX (Figure 5). C10Berb prevented rapid (18h) accumulation of lipofuscin induced by erastin or RSL3, indicating that this process was driven by mitoLPO (Figure 5). These data indicate that C10Berb inhibits ferroptosis due to its ability to block mitochondrial oxidative stress. In agreement with this finding, ferroptosis as well as mitoLPO and lipofuscin accumulation were blocked by the mitochondria-targeted antioxidant SkQ1 and methylene blue, which prevents ROS formation in the mitochondrial electron transport chain (Figure 1, Figure 4 and Figure 5). It is noteworthy that RSL3 engages multiple nodes of the cellular redox network: besides the commonly cited inhibition of GPX4 [39], recent chemoproteomic studies show that RSL3 covalently modifies several selenoproteins, including TXNRD1 and PRDX6, while sparing mitochondrial TXNRD2 [51]. Reports on whether RSL3 directly inhibits GPX4 remain inconsistent, with contradictory findings from Arner’s group and more recent work from Conrad and colleagues [52]. Although the origin of this discrepancy is unresolved, it underscores that RSL3 activates ferroptotic signaling through several parallel redox-active pathways, further highlighting the importance of mitochondrial oxidative stress as an upstream event counteracted by C10Berb.

Studies in various in vivo models have shown that the antioxidant effect of berberine may be mediated by activation of AMPK and Nrf2 [26]. Activation of AMPK by berberine, as well as by C10Berb, is mediated by efficient inhibition of complex I in the mitochondrial electron transport chain [35,36,42]. Pretreatment with the complex I inhibitor rotenone as well as the other known AMPK activators AICAR and 2-deoxyglucose did not protect against ferroptosis induced by erastin or RSL3 (Figure 3). Stimulation of Nrf2 using sulforaphane also failed to confer protection (Figure 3). These results suggest that neither AMPK nor Nrf2 activation underlies the anti-ferroptotic effect of C10Berb.

There are several studies demonstrating the strong iron binding capacity of berberine (see, for example, [20]). We did not analyze the possible role of iron chelation in the protective effect of C10Berb. On the other hand, we observed a direct antioxidant effect of C10Berb against lipid peroxidation in isolated cardiac mitochondria, where LPO was induced either by rotenone in the presence of iron and NAD-dependent respiratory substrates or by the lipophilic free radical initiator AMVN (Figure 6). The iron concentration in these experiments (50 μM FeSO_4_) was significantly higher than the effective concentration of C10Berb (0.3 μM), so the effect of iron chelation can be ruled out. Therefore, we propose that the antioxidant activity of C10Berb may be responsible for its protective effect.

Our results position C10Berb as a significantly more potent antioxidant and antiferroptotic agent compared to native berberine. Previous studies have described berberine as a weak antioxidant, with its biological effects often attributed to indirect pathways, such as activation of AMPK or Nrf2 [26]. However, its direct impact on mitochondrial oxidative stress has remained limited, in part due to its poor mitochondrial accumulation and low bioavailability. In contrast, our data show that C10Berb accumulates efficiently in mitochondria and directly suppresses key oxidative processes associated with ferroptosis.

This finding aligns with a growing interest in designing berberine derivatives that enhance specific bioactivities through improved pharmacokinetics or subcellular targeting. Although several analogues have been developed to increase berberine’s therapeutic potential, few have demonstrated a substantial improvement in direct antioxidant capacity, particularly within mitochondria. Our results therefore suggest that rational structural modification, specifically, promoting mitochondrial delivery, may be a viable strategy for transforming berberine from a moderately effective modulator of cellular metabolism into a direct and potent mitochondrial protector.

The therapeutic potential of berberine has been hampered by its poor water solubility and low gastrointestinal absorption. Developing berberine derivatives with improved bioavailability has thus emerged as a promising strategy in drug development. To date, hundreds of berberine analogues have been synthesized to overcome these limitations [53].

Various berberine derivatives have been shown to exhibit improved antidiabetic, anti-inflammatory, antimicrobial, and antiviral properties [54]. The most significant advances have been made in the synthesis and characterization of derivatives with improved DNA binding and anticancer activity [53]. 13-substituted benzo- and alkyl-berberines, among other derivatives, have been reported to exhibit superior activity against pathogenic fungi, mycobacteria, and human cancer cell lines [55]. The possible role of mitochondrial localization of berberine derivatives in enhancing their effects has not been analyzed; however, in an early study by Mikes and Dadák [56], the interaction of 13-substituted alkylberberines with isolated cardiac mitochondria was investigated. It was shown that the fluorescence quantum yield of the derivatives bound to mitochondria increases with increasing alkyl chain length from methyl to butyl. It is important to note that the generation of membrane potential on the inner mitochondrial membrane (negative inside) strongly increases the fluorescence of all berberine derivatives, whereas at the concentrations used (up to 3 μM) virtually no effect of the potential on the amount of bound dye was detected. These data indicate that the transmembrane potential causes deeper immersion of berberine derivatives into the hydrophobic region of the membrane, which increases the fluorescence quantum yield. The high fluorescence of 13-substituted decyl berberine (C10Berb) accumulated in the mitochondria of living cells (Figure 2) is fully consistent with this conclusion.

Only rare studies have been aimed at enhancing the antioxidant activity of berberine derivatives. It was shown that 9-N-(o-methylphenethyl)-substituted berberine has elevated antioxidant activity and also strongly inhibits acetylcholinesterase activity and β-amyloid aggregation [57]. Berberine 9-*O*-benzoic acid derivatives also have improved radical scavenging activity, which was revealed in several chemical assays [21]. Derivatives where chlorine-substituted piperazine moieties were linked to berberine at different positions were also reported to exhibit potent antioxidant effects [58,59].

Our results demonstrate that conjugation with a decyl chain markedly enhances mitochondrial accumulation, thereby transforming berberine into a highly effective mitochondrial antioxidant. This strategy aligns with emerging approaches in medicinal chemistry that emphasize organelle-specific targeting as a means of enhancing therapeutic efficacy. Mitochondria-targeted antioxidants such as SkQ1, MitoQ, and related derivatives have already demonstrated protective effects in models of cardiovascular, neurodegenerative, and metabolic diseases [60,61,62]. By combining the well-documented biological versatility of berberine with mitochondrial targeting, C10Berb represents a new class of compounds with the potential to bridge metabolic modulation and direct antioxidant defense.

Given the growing recognition of ferroptosis as a central contributor to ischemia–reperfusion injury, cardiomyopathy, neurodegeneration, and cancer therapy resistance, mitochondria-targeted berberine derivatives may hold considerable translational potential. Beyond ferroptosis, the ability of C10Berb to suppress mitochondrial oxidative stress suggests broader applications in conditions where mitochondrial dysfunction is a driver of pathology. Future work should assess the pharmacokinetics, tissue distribution, and in vivo efficacy of C10Berb in disease models, as well as explore structural optimization to balance potency, selectivity, and bioavailability.

## 5. Conclusions

In conclusion, this study establishes C10Berb as a prototype of a mitochondria-targeted berberine derivative with potent antioxidant and antiferroptotic activity. By overcoming the intrinsic limitations of berberine, C10Berb exemplifies how rational structural modification and subcellular targeting can transform a weakly active natural compound into a promising mitochondrial protector with potential therapeutic applications across a spectrum of ferroptosis-related diseases.

## Figures and Tables

**Figure 1 cells-14-01963-f001:**
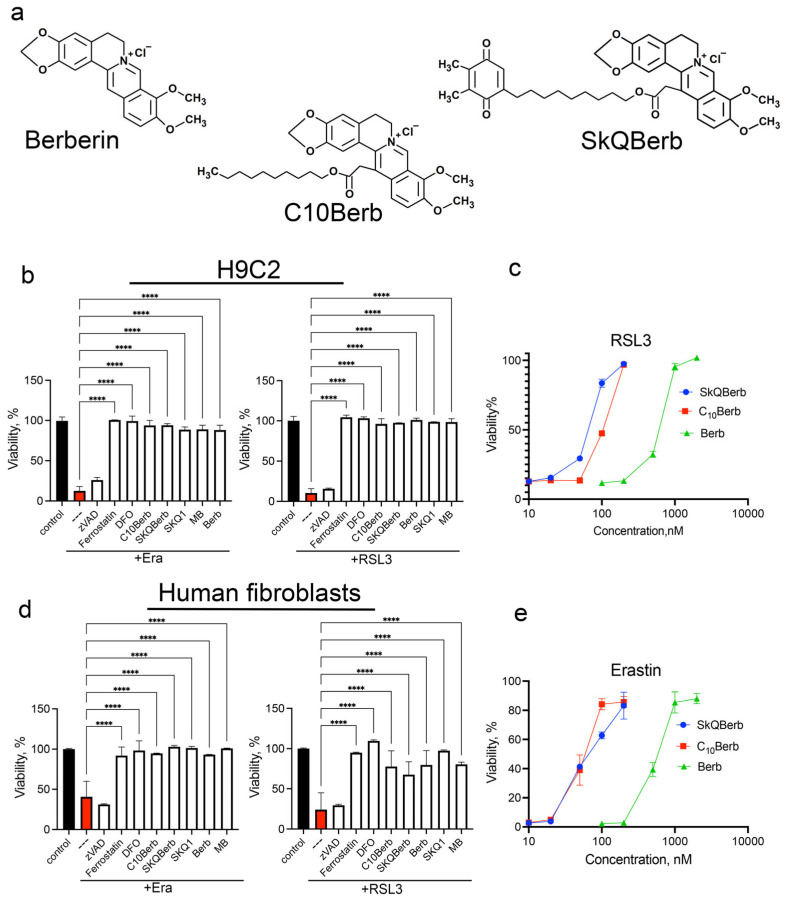
Berberine and its derivatives protected H9c2 cardiomyocytes and human diploid fibroblast from ferroptosis induced by erastin and RSL3. (**a**) Chemical structures of berberine, C10Berb and SkQBerb (**b**,**d**) Cells were incubated with 5 µM erastin or 200 nM RSL3 alone and in combination with 10 µM zVADfmk (zVAD), 10 µM ferrostatin-1, 100 µM deferoxamine (DFO), 100 nM C10Berb, 100 nM SkQBerb, 10 µM berberine (Berb), 100 nM SkQ1, or 250 nM methylene blue (MB) for 24 h. *p* < 0.0001 ****—significance of differences between erastin- or RSL3-treated samples and other samples (n = 6). (**c**,**e**) Protective efficacy of C10Berb, SKQBerb and berberine (Berb) against ferroptosis induced by RSL3 (200 nM) or erastin (5 µM) in H9c2 cardiomyocytes. Cell viability was measured after 24 h using the CellTiterBlue reagent.

**Figure 2 cells-14-01963-f002:**
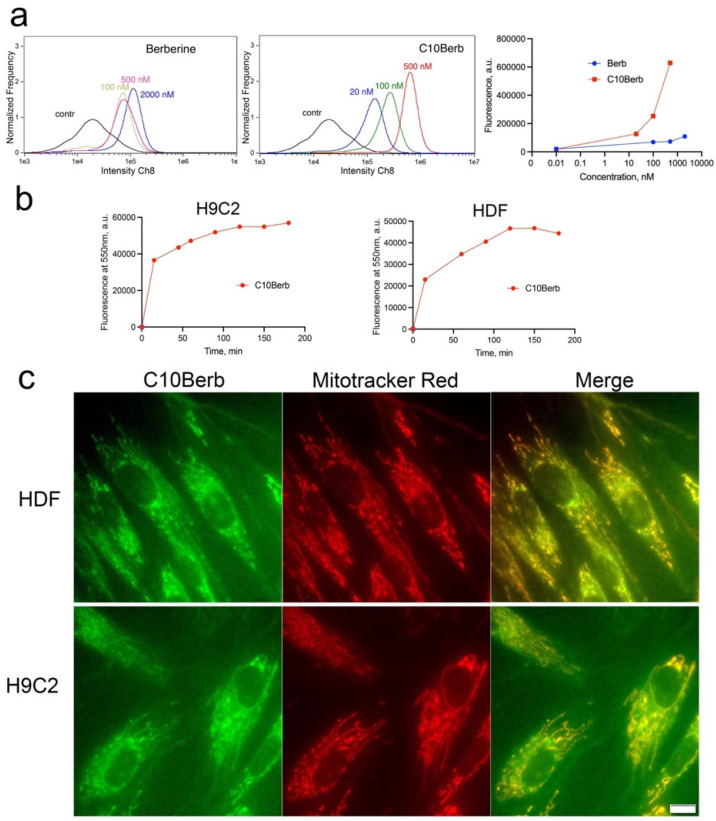
Accumulation of C10Berb and berberine in H9c2 cardiomyocytes and in human diploid fibroblasts (HDF). (**a**). H9c2 cardiomyocytes were incubated with C10Berb or berberine (Berb) at indicated concentrations for 1 h then washed with PBS and analyzed using an Amnis FlowSight flow cytometer. (**b**) Cells were incubated with 500 nM C10Berb for indicated periods, washed with PBS, suspended in PBS with 1% SDS and fluorescence was measured at 550 nm (excitation at 365 nm) using FluoroMax-3 spectrofluorometer. (**c**) Localization of C10Berb in H9C2 cells or in HDF was analyzed after incubation with 100 nM C10Berb for 1 h followed by staining of mitochondria with MitoTracker Red (100 nM, 15 min). Fluorescence was analyzed using a fluorescence microscope Olympus IX 83 (Tokyo, Japan). Bar—20 µm.

**Figure 3 cells-14-01963-f003:**
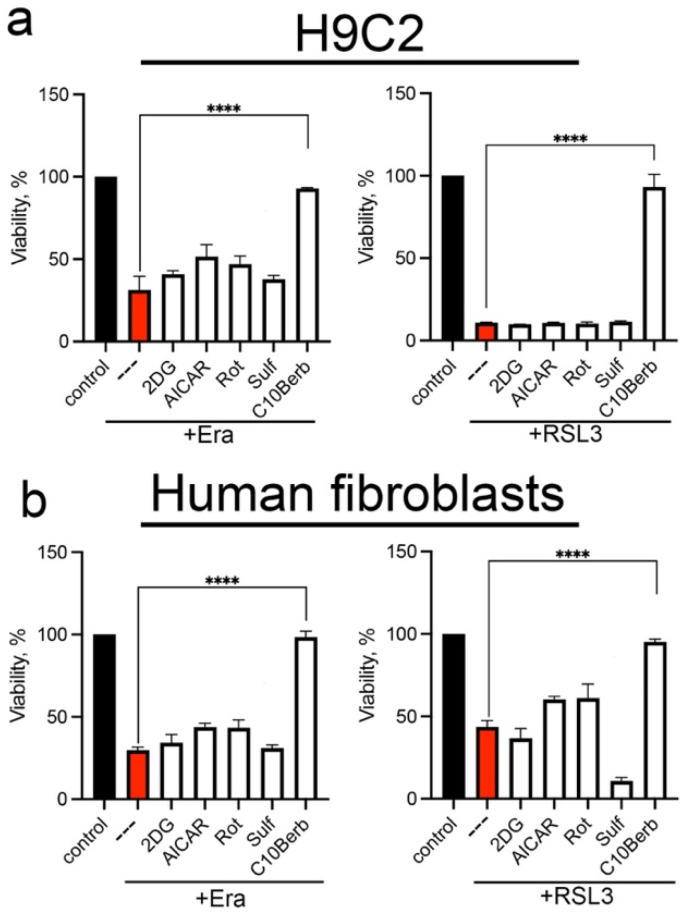
AMPK and Nrf2 inducers did not protect H9c2 cardiomyocytes (**a**) and human diploid fibroblasts (HDF) (**b**) from erastin- and RSL3-induced ferroptosis. Cells were incubated with 5 µM erastin or 200 nM RSL3 alone and in combination with 5 mM 2-deoxy-D-glucose (2DG), 0.5 mM AICAR, 2 µM rotenone (Rot), 10 µM sulforaphane (Sulf), and 100 nM C10Berb for 24 h. Cell viability was measured using the CellTiterBlue reagent. *p* < 0.0001 ****—significance of differences between samples (n = 5).

**Figure 4 cells-14-01963-f004:**
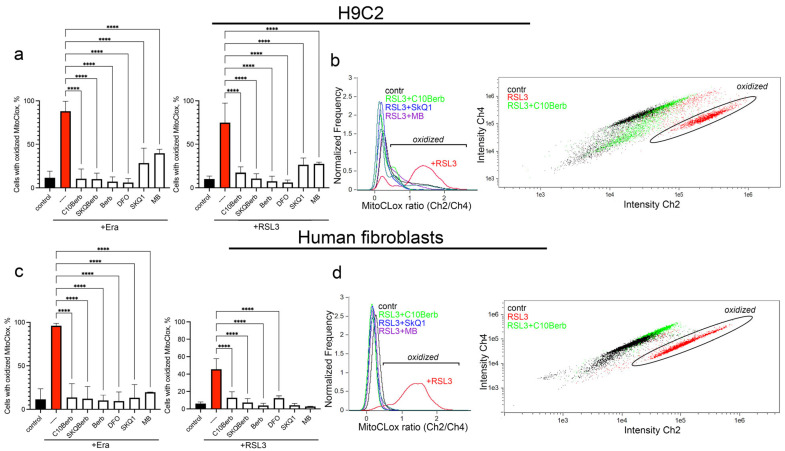
Erastin and RSL3 induce mitochondrial lipid peroxidation (LPO) in H9c2 cardiomyocytes and human fibroblasts. Cells were incubated with 5 µM erastin or 100 nM RSL3 alone and in combination with 100 nM C10Berb, 100 nM SkQBerb, 10 µM Berb, 100 µM DFO, 100 nM SkQ1, and 250 nM MB for 18 h. Mitochondria-targeted fluorescent ratiometric probe MitoCLox was added for 18 h. (**a**–**d**) Ratio of green/red fluorescence was measured and analyzed using gating procedure. *p* < 0.0001 ****—significance of the difference between samples treated with erastin or RSL3 and other samples (n = 6). (**b**,**d**) Representative histograms and dot plots illustrating the gating procedure.

**Figure 5 cells-14-01963-f005:**
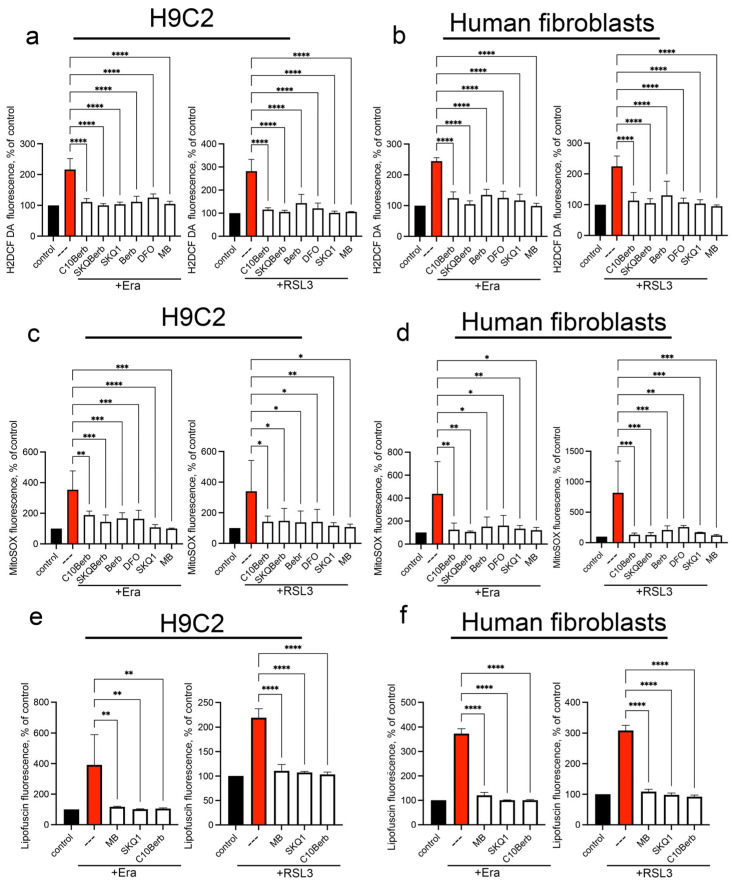
Erastin and RSL3 induce the production of mitochondrial superoxide anion and cytosolic ROS. Cells were incubated with 5 µM erastin or 100 nM RSL3 alone and in combination with 100 nM C10Berb, 100 nM SkQBerb, 10 µM Berb, 100 nM SkQ1, 100 µM DFO and 250 nM MB for 18 h. Cells were stained with 1.8 µM CM-H_2_DCFDA (**a**,**b**) or with 1 µM MitoSOX (**c**,**d**) for 30 min. Lipofuscin accumulation was measured by autofluorescence (**e**,**f**). The fluorescence of control samples was taken as 100% *p* < 0.0001 ****, *p* ≤ 0.001 ***, 0.05 ≥ *p* > 0.001 **, *p* < 0.05 *—significance of the difference between samples treated with erastin or RSL3 and other samples (n = 4). Representative flow cytometry plots corresponding to this figure are shown in Appendix A.

**Figure 6 cells-14-01963-f006:**
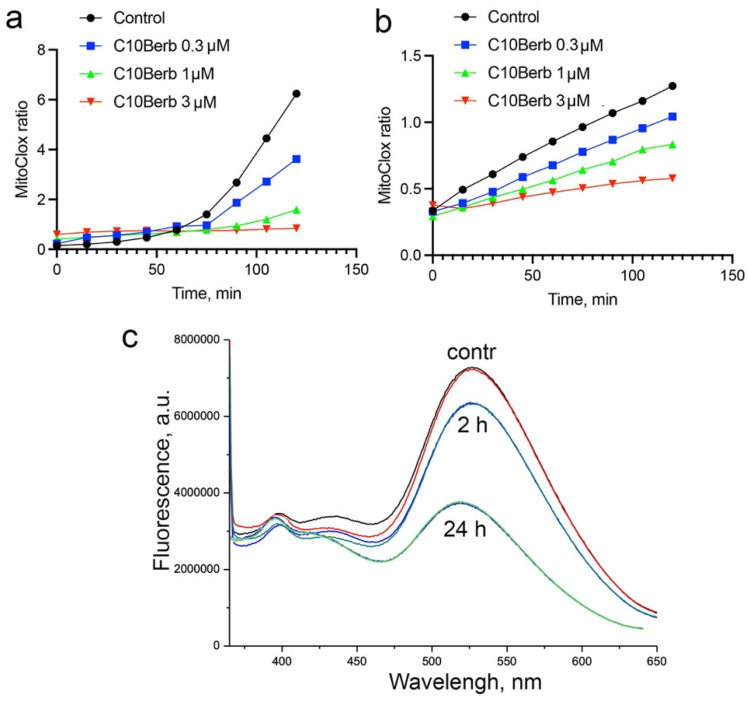
Effect of C10Berb on the dynamics of mitochondrial lipid peroxidation induced by rotenone in the presence of iron and NAD-dependent respiratory substrates (**a**), or by the lipophilic free radical initiator AMVN (**b**). Reduction in C10Berb by mitochondria during incubation with NAD-dependent respiratory substrates (**c**).

## Data Availability

The data that support the findings of this study are available from the corresponding author (Konstantin Lyamzaev) upon reasonable request.

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
