# Peer review of "13-Decyl Berberine Derivative Is a Novel Mitochondria-Targeted Antioxidant and a Potent Inhibitor of Ferroptosis"

_cells, 2025, doi:10.3390/cells14241963_

Round 1
Reviewer 1 Report
Comments and Suggestions for Authors
In this manuscript, the authors mainly investigate the mitochondria-targeted antioxidant activity of the berberine derivative C10Berb, as well as its potential role as a ferroptosis inhibitor. Throughout the paper, the authors convincingly demonstrate that C10Berb specifically localizes to mitochondria and possesses both antioxidant and anti-ferroptotic activities. The study is highly original, and the experimental strategies adopted are generally appropriate.
However, after a thorough review, I have identified two major issues that require further clarification and experimental verification by the authors.
- Comparison between C10Berb and SKQ derivatives, and the need for a berberine control group
Both C10Berb and SKQBerb are berberine derivatives. According to the experimental results, C10Berb exhibits properties similar to those of SKQBerb or SKQ1. Structurally, however, C10Berb does not contain the SKQ1 moiety. These results suggest that the addition of a long hydrocarbon chain (C10) could alter the characteristics of C10Berb, endowing it with SKQ1-like functionality. Moreover, since SKQBerb and SKQ1 display similar effects (without any additive effect), this implies that they may share the same intracellular targets (or substrates), thereby influencing cellular activity.
My question is: Should berberine itself be included as a control in some of these experiments? Although Figure 2a shows that intracellular levels of berberine are relatively low, it may still be worthwhile to test a higher concentration (e.g., 1 μM) to determine its subcellular localization and effects on mitochondrial lipid peroxidation, among other parameters. - Verification of AMPK activation by C10Berb
In the Introduction, Results (section 3.2), and Discussion, the authors state that C10Berb exerts its antioxidant and anti-ferroptotic effects independently of cytoplasmic activation pathways (such as AMPK activation). This assumption is based on the premise that C10Berb is capable of activating AMPK; however, no supporting data are presented. Although several references (Refs. 36, 37, 43, 26, 27, etc.) are cited, none directly demonstrate that C10Berb (as distinct from other berberine derivatives) activates AMPK.
To substantiate this claim, the authors should provide experimental data showing p-AMPK (Thr172), p-ACC (S79), and Nrf2 nuclear translocation.
As noted in the previous point, I also question whether C10Berb might have undergone a change in functional characteristics (i.e., from berberine-like to SKQ1-like). If this is the case, the rationale for the experiments described in Section 3.2 would be undermined. Therefore, I strongly recommend that the authors include experimental results verifying whether C10Berb indeed activates the AMPK pathway.
In addition, several other points require further clarification or improvement. For example, the authors should include the flow cytometry plots corresponding to Figure 5, and in lines 290–322, the figure references should be stated more precisely (e.g., in line 297, instead of simply “Figure 5,” it should specify “Figure 5a and 5b,” etc.).
Finally, I am also curious whether C10Berb occurs naturally in any known biological system, or if it is currently only synthetically produced.
Author Response
Comment 1. Comparison between C10Berb and SKQ derivatives, and the need for a berberine control group.
Both C10Berb and SKQBerb are berberine derivatives. According to the experimental results, C10Berb exhibits properties similar to those of SKQBerb or SKQ1. Structurally, however, C10Berb does not contain the SKQ1 moiety. These results suggest that the addition of a long hydrocarbon chain (C10) could alter the characteristics of C10Berb, endowing it with SKQ1-like functionality. Moreover, since SKQBerb and SKQ1 display similar effects (without any additive effect), this implies that they may share the same intracellular targets (or substrates), thereby influencing cellular activity.
My question is: Should berberine itself be included as a control in some of these experiments? Although Figure 2a shows that intracellular levels of berberine are relatively low, it may still be worthwhile to test a higher concentration (e.g., 1 μM) to determine its subcellular localization and effects on mitochondrial lipid peroxidation, among other parameters.
Response 1.
Thank you for this comment. We conducted additional experiments with berberine. The results are presented in Figures 3–5. These data show that berberine at a dose 100 times higher than C10Berb not only protects against ferroptosis but also inhibits mitochondrial lipid peroxidation and the accumulation of both mitochondrial and cytosolic ROS. These results indicate that the protective effect of berberine is mediated by its antioxidant activity. Berberine accumulation in cells at these high concentrations (up to 10 μM) was significantly lower than that of C10Berb at 100 nM. Even at 10 μM, we did not observe any selective accumulation of berberine in mitochondria (Fig S3). We concluded that the lower protective efficacy of berberine compared to C10Berb is due to its lower cellular content and lack of selective accumulation in mitochondria.
Comment 2. Verification of AMPK activation by C10Berb.
In the Introduction, Results (section 3.2), and Discussion, the authors state that C10Berb exerts its antioxidant and anti-ferroptotic effects independently of cytoplasmic activation pathways (such as AMPK activation). This assumption is based on the premise that C10Berb is capable of activating AMPK; however, no supporting data are presented. Although several references (Refs. 36, 37, 43, 26, 27, etc.) are cited, none directly demonstrate that C10Berb (as distinct from other berberine derivatives) activates AMPK.
To substantiate this claim, the authors should provide experimental data showing p-AMPK (Thr172), p-ACC (S79), and Nrf2 nuclear translocation.
As noted in the previous point, I also question whether C10Berb might have undergone a change in functional characteristics (i.e., from berberine-like to SKQ1-like). If this is the case, the rationale for the experiments described in Section 3.2 would be undermined. Therefore, I strongly recommend that the authors include experimental results verifying whether C10Berb indeed activates the AMPK pathway.
Response 2
We conducted additional experiments analyzing the potential activation of AMPK by C10Berb and berberine. SkQ1 was included in these analyses for comparison. In these experiments, we did not observe significant accumulation of p-AMPK (Thr172), indicating a lack of AMPK activation. These data contradict numerous reports of AMPK activation by berberine, and we have no explanation for this discrepancy. It should be emphasized that these data are not critical to our logic (based on the lack of effects of various AMPK activators), but they support the conclusion that the protective effect of C10Berb is independent of AMPK.
To analyze the possible role of NRF2 activation in the protective effect of C10Berb, we measured NRF2 protein accumulation (Western blotting) and heme oxygenase-1 expression (RT-PCR), which are targets of NRF2-dependent gene expression. Berberine (10 μM), but not C10Berb (100 nM) or SkQ1 (100 nM), was shown to significantly activate NRF2 (Fig.S1 and S2). These data indicate that NRF2 activation is not responsible for the protective effect of C1Berb and SkQ1, which is consistent with the lack of protection observed with sulforaphane, a potent NRF2 activator. A role for NRF2 activation in the protective and antioxidant effects of berberine cannot be completely ruled out, but it seems questionable given the lack of effect of sulforaphane.
The results of these experiments were include in the revised version of our manuscript as a supplement.
Comment 3
In addition, several other points require further clarification or improvement. For example, the authors should include the flow cytometry plots corresponding to Figure 5, and in lines 290–322, the figure references should be stated more precisely (e.g., in line 297, instead of simply “Figure 5,” it should specify “Figure 5a and 5b,” etc.).
Response 3.
Figure 5 contains a significant amount of information on two cell types, two ROS-sensitive dyes, and autofluorescence data reporting on lipofuscin content. Representative flow cytometry plots corresponding to Figure 5 are presented below. These plots demonstrate a normal distribution of fluorescence across the cell population in all cases, so Figure 5 presents the mean values. Flow cytometry plots have not been included in the revised version to avoid overloading.
Comment 4
Finally, I am also curious whether C10Berb occurs naturally in any known biological system, or if it is currently only synthetically produced.
Response 4
We have found no information about the presence of C10Berb or any similar berberine derivatives in natural products or living beings.

Reviewer 2 Report
Comments and Suggestions for Authors
The manuscript is focused on a berberine derivative that shows mitochnodrial accumulation while preventing ferroptosis induced by two molecules. Missing in the manuscript is a clear conclusion identifying the berberine derivative as an antioxidant. Along with that, experiments are missing to rule out ability of the derivative to chelate free iron or serve as agonist/inducer of the proteins affected by erastin or RSL3.
Detailed comments:
- The inhibitors of ferroptosis used in the manuscript are iron chelators. Yet there are no experiments showing berberine or its derivatives are capable of the same.
- RSL3 has been shown to inhibit TXNRD1 and not GPX4 - Cheff DM et al. Redox Biol 2023. It is still an inducer of ferroptosis but the reference should be included in Results when describing activity of RSL3 and considered elsewhere in the Results (Figure 4).
- Both of the ferroptosis inducers affect cysteine or selenocysteine availability. What if the berberine derivative induces either or both proteins expression?
- It would be helpful to rule out direct radical scavenging by berberine derivative in a non-cellular model.
Author Response
Comment 1.
The inhibitors of ferroptosis used in the manuscript are iron chelators. Yet there are no experiments showing berberine or its derivatives are capable of the same.
Response 1.
There are several studies demonstrating the strong iron binding capacity of berberine (see, for example, Shirwaikar A, Shirwaikar A, Rajendran K, Punitha IS. In vitro antioxidant studies on the benzyl tetra isoquinoline alkaloid berberine. Biol Pharm Bull. 2006;29(9):1906-1910. doi:10.1248/bpb.29.1906). We did not analyze the possible role of iron chelation in the protective effect of C10Berb and have indicated this in the text. On the other hand, we observed a direct antioxidant effect of C10Berb against lipid peroxidation in isolated mitochondria (Fig. 6). The iron concentration in these experiments (50 μM FeSOâ‚„) was significantly higher than the effective concentration of C10Berb (0.3 μM), so the effect of iron chelation can be ruled out. Therefore, we propose that the antioxidant activity of C10Berb may be responsible for its protective effect.
Comment 2.
RSL3 has been shown to inhibit TXNRD1 and not GPX4 - Cheff DM et al. Redox Biol 2023. It is still an inducer of ferroptosis but the reference should be included in Results when describing activity of RSL3 and considered elsewhere in the Results (Figure 4).
Response 2.
Indeed, it is well known that RSL3 targets almost all selenoproteins, with the exception of TXNRD2 (for the most recent publication, see DeAngelo SL, Zhao L, Dziechciarz S, et al. Recharacterization of the Tumor Suppressive Mechanism of RSL3 identifies the Selenoproteome as a Druggable Pathway in Colorectal Cancer. Cancer Res. 2025;85(15):2788-2804. doi:10.1158/0008-5472.CAN-24-3478). In particular, RSL3 inhibits TXNRD1 and PRDX6, which are involved in protecting cells from ferroptosis. GPX4 has been shown to be a critical target of RSL3 for ferroptosis (Yang WS, SriRamaratnam R, Welsch ME, et al. Regulation of ferroptotic cancer cell death by GPX4. Cell. 2014;156(1-2):317-331. doi:10.1016/j.cell.2013.12.010). In contrast, as pointed out by the distinguished Reviewer, Elias Arner and colleagues reported that RSL3 does not inhibit GPX4. These data completely contradict a later study by M. Conrad and colleagues (Nakamura T, Ito J, Mourão ASD, et al. A tangible method to assess native ferroptosis suppressor activity. Cell Rep Methods. 2024;4(3):100710. doi:10.1016/j.crmeth.2024.100710). The reason for this discrepancy is unclear. We have briefly addressed this discrepancy and include the references in the revised version.
Comment 3
Both of the ferroptosis inducers affect cysteine or selenocysteine availability. What if the berberine derivative induces either or both proteins expression?
Response 3
Thank you for this interesting comment. We found no evidence in the literature of an effect of berberine or its derivatives on selenoprotein expression or selenocysteine availability. This possibility may be a promising avenue for future research.
Comment 4
It would be helpful to rule out direct radical scavenging by berberine derivative in a non-cellular model.
Response 4
Data confirming the direct radical scavenging activity of C10Berb are presented in Figure 6. We observed that C10Berb effectively prevented lipid peroxidation in isolated mitochondria. This effect was observed both when lipid peroxidation was induced by mitochondrial ROS production and when using an artificial free radical scavenger.
Round 2
Reviewer 1 Report
Comments and Suggestions for Authors
The authors have addressed most of my previous questions in the revised manuscript. However, regarding my earlier suggestion about Figure 5, although the authors mentioned the need to avoid overloading the manuscript, I still believe that the flow cytometry data should be presented. Even if it cannot be included in the main text, it could be provided as a supplementary figure. Simply placing the relevant graphics in the author's reply folder is not enough.
Author Response
Comment 1
The authors have addressed most of my previous questions in the revised manuscript. However, regarding my earlier suggestion about Figure 5, although the authors mentioned the need to avoid overloading the manuscript, I still believe that the flow cytometry data should be presented. Even if it cannot be included in the main text, it could be provided as a supplementary figure. Simply placing the relevant graphics in the author's reply folder is not enough.
Response 1
Thank you for reviewing our manuscript and for your comments regarding Figure 5. We recognize the importance of presenting flow cytometry data. In response to your comment, we have added an additional figure containing typical flow cytometry graphs to the supplementary materials (now designated as Figure S4). A corresponding reference to this additional figure has also been added to the caption of Figure 5 in the main text.